# The Influence of Internal Intermittency, Large Scale Inhomogeneity, and Impeller Type on Drop Size Distribution in Turbulent Liquid-Liquid Dispersions

**DOI:** 10.3390/e21040340

**Published:** 2019-03-28

**Authors:** Wioletta Podgórska

**Affiliations:** Faculty of Chemical and Process Engineering, Warsaw University of Technology, 00-645 Warsaw, Poland; wioletta.podgorska@pw.edu.pl

**Keywords:** drop breakage, drop coalescence, local intermittency, turbulent flow, population balance equation, high efficiency impeller, Rushton turbine

## Abstract

The influence of the impeller type on drop size distribution (DSD) in turbulent liquid-liquid dispersion is considered in this paper. The effects of the application of two impellers, high power number, high shear impeller (six blade Rushton turbine, RT) and three blade low power number, and a high efficiency impeller (HE3) are compared. Large-scale and fine-scale inhomogeneity are taken into account. The flow field and the properties of the turbulence (energy dissipation rate and integral scale of turbulence) in the agitated vessel are determined using the k-ε model. The intermittency of turbulence is taken into account in droplet breakage and coalescence models by using multifractal formalism. The solution of the population balance equation for lean dispersions (when the only breakage takes place) with a dispersed phase of low viscosity (pure system or system containing surfactant), as well as high viscosity, show that at the same power input per unit mass HE3 impeller produces much smaller droplets. In the case of fast coalescence (low dispersed phase viscosity, no surfactant), the model predicts similar droplets generated by both impellers. In the case of a dispersed phase of high viscosity, when the mobility of the drop surface is reduced, HE3 produces slightly smaller droplets.

## 1. Introduction

Liquid-liquid dispersions in a turbulent flow are common in many applications in chemical, petroleum, pharmaceutical, and food industries. Processes involving liquid-liquid dispersions include suspension polymerization, extraction, and heterogeneous reactions. The rate of a heterogeneous chemical reaction is often controlled by mass transfer. Mass transfer is also the base of the extraction process. The efficiency of mass transfer strongly depends on the interfacial area determined by drop size distribution, which in turn is controlled by drop breakage and coalescence processes. Drop size distribution also determines the quality of the product obtained in suspension polymerization. Droplet breakage, which is a short-duration process, i.e., the process characterized by time scales smaller than time constants of related turbulent events, can be strongly influenced by internal intermittency (also called local or fine-scale intermittency) [1,2]. Internal intermittency also affects the coalescence process. Internal intermittency results from vortex stretching, which leads to the formation of regions of space characterized by high vorticity surrounded by nearly irrotational fluid. Small scale intermittency can be deduced from probability distribution functions of velocity gradients and differences [3,4]. From the distribution of the velocity derivatives, it is evident that the energy associated with large wave numbers (small length scales) is very unevenly distributed. Dissipation associated with increasing wavenumbers becomes increasingly concentrated in small regions [5,6]. It means that there are regions and periods of activity and quiescence. This spotty distribution in time and space manifests in an anomalous scaling of fluctuating quantities. Two scaling laws of special interest are those for a velocity increment over a distance, *r*: (1)$$\langle {\left(\delta \;u\left(r\right)\right)}^{p}\rangle \sim r^{\zeta_{p}},$$
and for energy dissipation, *ε*, averaged over a ball of a size, *r*: (2)$$\langle \varepsilon_{r}^{p}\rangle \sim r^{\tau_{p}}.$$

The exponent of the structure function, *ζ_p_*, differs from *p*/3 predicted by Kolmogorov theory and the discrepancy between *ζ_p_* and *p*/3 increases with increasing *p*. For positive *p*, the exponents in Equations (1) and (2) are related by *ζ_p_* = *p*/3 + *τ*_*p*/3_ [7,8]. The intermittent character of turbulence can be modeled using multifractal formalism [4,8]. There are theoretical arguments for this formalism related to the nonlinear character of Navier-Stokes equations. There exists a strange attractor for Navier-Stokes equation (N-S) and solutions attracted to the strange attractor correspond to the turbulence. Instantaneous realization of the flow or any instantaneous solution of an N-S equation can be treated as an object consisting of various objects related to fractal sets embedded in physical space. The N-S equations in the zero viscosity limit are invariant under the following group of rescaling transformations: *x_i_*’ = *λx_i_*, *u_i_*’ = *λ^h^u_i_*, and *t*’ = *λ*^1*-h*^*t*, provided that < *η* > < *r*, *r*’ < *L* and *L* >> < *η* >, where $r=\sqrt{x_{i}^{2}}$. *L* is the integral scale of turbulence, < *η* > is the Kolmogorov microscale, and *h* is a scaling exponent. When the viscous term is neglected at high Reynolds numbers, there are infinitely many scaling groups, labeled by their scaling exponent, *h*, which can be any real number [8]. When one considers energy dissipation, then $\varepsilon_{r}/\varepsilon_{L}\propto {\left(r/L\right)}^{\alpha -d_{s}}$, where $\varepsilon_{L}$ is the average of ε over a box of a size, *L*; *α* is a scaling exponent (also called a multifractal exponent or singularity strength); and *d_s_* is the space dimension. Scaling exponents for velocity, *h*, and for dissipation, *α*, are related by *h* = *α*/3 [4,8]. The transformation for dynamic pressure, *p*, can be neglected because the pressure can be eliminated from the Navier-Stokes equation [8]. However, in turbulent flow, the breakage of droplets with a size from the inertial subrange results from dynamic pressure fluctuations, thus the scaling law for pressure is of interest. The pressure transforms as $u_{i}^{2}$, or scales as $p'=\lambda^{2\alpha /3}p$ and the local normal pressure stresses in the inertial subrange acting on droplets of a size, *d*, are [2]: (3)$$p\left(d,\alpha \right)=C_{p}\rho_{C}{\left[\langle \varepsilon \rangle d\right]}^{2/3}{\left(\frac{d}{L}\right)}^{\frac{2}{3}\left(\alpha -1\right)}.$$
The velocity increment over a distance, *r*, is:(4)$$u_{r}={\left[\langle \varepsilon \rangle r\right]}^{1/3}{\left(\frac{d}{L}\right)}^{\frac{\alpha -1}{3}}.$$
At pure breakage (i.e., when coalescence is negligible), the maximum stable drop size, $d_{\max}$ (for dispersed phase of low viscosity), results from the balance of pressure stresses given by Equation (3) and shape restoring stresses given by *σ*/*d*, where *σ* is an interfacial tension [2]:(5)$$d_{\max}=C_{x}^{\frac{5}{3+2\alpha }}L{\left(\frac{\sigma }{\rho_{C}{\langle \varepsilon \rangle }^{2/3}L^{5/3}}\right)}^{\frac{3}{3+2\alpha }}.$$
For viscous drops, the additional stabilizing stress (viscous stress) should be taken into account, thus *d*_max_ is given by [2]: (6)$$d_{\max}=C_{x}\frac{\sigma^{0.6}}{\rho_{C}^{0.6}{\langle \varepsilon \rangle }^{0.4}}{\left(\frac{d_{\max}}{L}\right)}^{0.4\left(1-\alpha \right)}{\left[1+\beta_{\mu }\mu_{D}{\left(\frac{d_{\max}}{L}\right)}^{\frac{\alpha -1}{3}}\frac{{\langle \varepsilon \rangle }^{1/3}d_{\max}^{1/3}}{\sigma }\right]}^{0.6}.$$
where $\rho_{C}$ is a continuous phase density, $\mu_{D}$ is a dispersed phase viscosity, and $C_{x}$ and $\beta_{\mu }$ are constants. Equations (5) and (6) do not give any reference to time. Drop size evolution in time can be predicted by solving the population balance equation with suitable breakage and coalescence models.

Models of breakage and coalescence are usually based on a classical Kolmogorov theory of turbulence that neglects intermittency. One of the first and most popular breakage models was proposed by Coulaloglou and Tavlarides [9]. The authors assumed that the droplet would be broken if the kinetic energy transmitted from eddies to the drop is larger than the drop surface energy. The fraction of eddies interacting with the droplet that have a kinetic energy larger than the surface energy is equal to the fraction of eddies that have velocities larger than the corresponding fluctuating velocity. It was assumed that only energies associated with velocity fluctuations of a scale smaller than the drop diameter tend to disperse the drop. A Gaussian distribution of turbulent velocity was assumed. Chatzi and Lee [10] and Chatzi et al. [11] assumed that the probability density of the kinetic energy of eddies is described by three-dimensional Maxwell distribution. Narsimhan et al. [12] treated droplets as one-dimensional simple harmonic oscillators. According to their model the oscillations of a drop are induced by the arrival of eddies of different scales and frequencies, and the number of arriving eddies is assumed to be a Poisson process. Konno et al. [13] assumed that breakage is caused by nonisotropic turbulence inside the impeller-disc edge and isotropic turbulence outside the impeller-disc edge. The breakage frequency in the region of isotropic turbulence was derived by using assumptions similar to those proposed by Coulaloglou and Tavlarides [9], but the probability density function of relative velocity was represented by Maxwell distribution. In the nonisotropic turbulent region, regularity in the direction of droplet elongation was observed. Therefore, breakage frequency was derived under the assumption that large energy-containing eddies are responsible for drop deformation and disruption. Martinez-Bazan et al. [14] based their model on a purely kinematic idea. They postulated that the acceleration of the fluid particle interface during deformation is proportional to the difference between the deformation and restoring stresses. All the models were derived for droplets of a size corresponding to the inertial subrange of scales. There is a group of breakage models based on a concept of collisions between droplets and eddies [15,16,17]. In recent years, these models, which were also formulated for the inertial subrange, were extended by using a wide energy spectrum [18,19,20,21]. The important question that appears when a breakage model is formulated is whether the droplet is broken by eddies smaller than the droplet, eddies of a size comparable to the drop diameter, or eddies larger than the droplet. According to Hinze [22], the droplet is disrupted by eddies of the same scale. Larger eddies only convey the drop, while smaller eddies are too weak to disperse the drop. In Coulaloglou and Tavlarides’ model [9], eddies smaller than the drop are responsible for breakage, while Andersson and Andersson [23,24] argue that eddies of a size approximately equal to and up to three times larger than the drop are responsible for dispersion. There are also breakage models taking into account the increased viscosity of the dispersed phase [25,26]. These models were further modified by Maaβ and Kraume [27], who assumed that the two mechanisms of breakage operate simultaneously (breakage induced by pressure fluctuations and breakage induced by two-dimensional elongational flow). All these breakage models neglect intermittency. However, as was discussed earlier, the local intermittency can have a profound effect on breakage and a noticeable effect on coalescence. Multifractal breakage models taking into account internal intermittency [2,28] allow the scale effect on the drop size to be explained; they explain the drift of the exponent on the Weber number from −0.6 to −0.93 in dimensionless relation for a maximum stable drop:(7)$$\frac{d_{\max}}{D}\propto We^{-0.6\left(\frac{1}{1-0.4\left(1-\alpha \right)}\right)}.$$
when the multifractal exponent changes from 1 to the infimum value of 0.12. They also explain the slow drift of transient drop size distributions at long agitation times. In these models, the concept of drop-eddy collision is not used. Models for droplets smaller than the Kolmogorov scale were also formulated using multifractal formalism [2]. Multifractal breakage and coalescence models allow proper predictions to be made of the changes of the drop size distribution both for short and long agitation times [2,28,29,30,31,32,33].

The coalescence process can be considered as an interaction triangle consisting of the continuous phase flow and two fluid particles. A continuous phase flow can be split into the external flow responsible for droplet collisions and the internal flow responsible for film drainage between colliding droplets [34]. The frequency of collisions of droplets of a size from the inertial subrange is based on the relative drop velocity, which is calculated as the characteristic velocity variation in the basic flow over a distance, *d* [35]. Another possible assumption is that colliding drops take the velocity of an eddy of the same size [9,15]. The efficiency of collisions depends on the drop surface mobility, drop size, surface deformation, etc. In pure liquid-liquid systems, partially mobile interfaces can be assumed [29,30,34]. In many models, immobilized interfaces are assumed [9,36,37]. Immobilization may be caused by the surfactant presence or high dispersed phase viscosity. A mobility parameter dependent on the viscosity ratio, $\mu_{D}/\mu_{C}$, can also be introduced to model the coalescence of droplets of a relatively high viscosity [15,31,32,38].

Immiscible liquids are often contacted in stirred vessels. Therefore, the geometry of the tank and the type of the used impeller are of great importance for producing a desired drop size distribution. Impellers can be classified as producing shear or flow. Radial disc turbines, like a Rushton turbine, commonly used for liquid-liquid systems, produce strong radial flow as well as intense turbulence. They can produce a high interfacial area. Hydrofoil impellers, such as Lightnin A310 or Chemineer HE3, produce axial or mixed flow and are especially good for systems differing in the density of the continuous and disperse phase. They have blades mounted at a shallow angle to reduce drag at the leading edges, and provide intensive axial flow with small power requirements. They are able to achieve a suspended state at a lower rotational speed than disc turbines. Therefore, they are particularly suitable for solid-liquid systems [39,40]. However, it was shown that the low power number high flow agitators, like HE3, can be used for liquid-liquid dispersions and produce smaller droplets than high power numbers, high shear agitators at the same power input per unit mass (i.e., the same average energy dissipation rate in the tank, $\bar{\langle \varepsilon \rangle }$) [41,42]. Therefore, in this paper, the influence of the impeller type on drop size distribution is presented. Two types of impellers are considered: Six-blade Rushton turbine (RT) and three-blade high efficiency impeller (HE3). The distribution of the locally averaged properties of the turbulence (including energy dissipation rate, ε, and integral scale of turbulence, *L*) are determined using the computational fluid dynamics CFD method. The distribution of these properties in the stirred tank affects the drop breakage and coalescence rates. 

Both processes (breakage and coalescence) are taken into account in this paper. Breakage takes place in practice only in the zone of the highest energy dissipation rate (impeller zone). The zone in the agitated tank where coalescence is privileged depends on the drop deformation in the contact area and on the mobility of the drop interfaces. The rates of both the breakage and coalescence depend on the mean power input per unit mass, and on a strong local and instantaneous variability of the energy dissipation rate related to the internal intermittency. Multifractal formalism was applied to model fine-scale intermittency.

## 2. Breakage and Coalescence Models

The time evolution of drop size distribution in a stirred tank is predicted by solving the population balance equation. A population of droplets of a volume, *υ*, and diameter, *d*, (*υ* = π*d*^3^/6) from the inertial subrange of turbulence is considered. The macroscopic population balance equation (averaged in the external phase space) formulated in the volume domain (for one internal coordinate corresponding to the drop volume) for chemically equilibrated liquid-liquid dispersion (with no mass transport) and batch operation is given by: (8)$$\begin{array}{ll} \frac{\partial n\left(\upsilon ,t\right)}{\partial t} & =\frac{1}{2}\int_{0}^{\upsilon }h\left(\upsilon -\upsilon^{\prime },\upsilon^{\prime }\right)\lambda \left(\upsilon -\upsilon^{\prime },\upsilon^{\prime }\right)n\left(\upsilon -\upsilon^{\prime }\right)n\left(\upsilon^{\prime }\right)d\upsilon^{\prime }-n\left(\upsilon ,t\right)\int_{0}^{\infty }h\left(\upsilon ,\upsilon^{\prime }\right)\lambda \left(\upsilon ,\upsilon^{\prime }\right)n\left(\upsilon^{\prime },t\right)d\upsilon^{\prime }\\ & +\int_{\upsilon }^{\infty }\beta \left(\upsilon ,\upsilon^{\prime }\right)\nu \left(\upsilon^{\prime }\right)g\left(\upsilon^{\prime }\right)n\left(\upsilon^{\prime },t\right)d\upsilon^{\prime }-g\left(\upsilon \right)n\left(\upsilon ,t\right) \end{array}$$
where *n*(*υ*,*t*) is the number density of drops of a volume, *υ*, at time, *t* (m^−6^). The drop breakage rate *g*(*d*) = *g*(*υ*) (s^−1^) in intermittent turbulent flow was developed by summing up the contributions to the break-up frequency from all vigorous eddies [2]:(9)$$g\left(d\right)=\int_{\alpha_{\min}}^{\alpha_{x}}g\left(\alpha ,d\right)P\left(\alpha \right)d\alpha =C_{g}\sqrt{\ln\left(\frac{L}{d}\right)}\frac{{\langle \varepsilon \rangle }^{1/3}}{d^{2/3}}{\int_{\alpha_{\min}}^{\alpha_{x}}\left(\frac{d}{L}\right)}^{\frac{\left(\alpha +2-3f\left(\alpha \right)\right)}{3}}d\alpha .$$
*P*(*α*) is a probability density for α in a box of a length, *r*; *g*(*α*,*d*) is the characteristic frequency of eddies of a size, *d*, labeled by a scaling exponent, *α*. Vigorous eddies that can disperse the drop are characterized by a multifractal exponent, *α*, from the range (*α*_min_, *α*_x_). The most vigorous eddies are characterized by *α*_min_. This value is difficult to measure and entails the extrapolation procedure. It was approximated for tails of the probability density of dissipation in boxes of a size *r*, *E_r_*, normalized by the overall dissipation, *E_t_*. The tails of distribution of (*E_r_*/*E_t_*) were found to be of the square-root exponential type and *α_min_* = 0.12 [4]. The upper bound of the integral in Equation (9), *α_x_*, results from the balance of stresses acting on the droplet and characterizes the weakest eddies that can disperse the drop [2,28]. The multifractal spectrum, *f*(*α*), is for practical reasons approximated by a polynomial [2] fitted to the experimental spectrum [4]. Thus, *f*(*α*) is given by:(10)$$f\left(\alpha \right)=a+b\alpha +c\alpha^{2}+d\alpha^{3}+e\alpha^{4}+f\alpha^{5}+g\alpha^{6}+h\alpha^{7}+i\alpha^{8},$$
where *a* = −3.4948, *b* = 18.721473, *c* = −55.918539, *d* = 120.90274, *e* = −162.54397, *f* = 131.51049, *g* = −62.572242, *h* = 16.1, and *i* = −1.7264619. The constant, *C_g_*, in Equation (9) is equal to *C_g_* = 0.0035. Depending on the liquid-liquid system, different stresses act on droplets. When the dispersed phase viscosity is low, the only stabilizing stress that opposes the disruptive turbulent stress given by Equation (3) is the shape restoring stress associated with interfacial tension, *σ*, $\tau_{\sigma }\propto \sigma /d$. Thus, the multifractal exponent, $\alpha_{x}$, resulting from the stress balance is given by:(11)$$\alpha_{x}=\frac{2.5\ln[(L{\langle \varepsilon \rangle }^{0.4}\rho_{C}^{0.6})/(C_{x}\sigma^{0.6})]}{\ln\left(L/d\right)}-1.5,$$
where the constant is *C_x_* = 0.23. High dispersed phase viscosity, $\mu_{D}$, increases the stabilizing effect. The viscous stress inside the drop is generated when a drop deforms. Thus, there are viscous and interfacial tension stresses that oppose the turbulent disruptive stress [2]. The droplet must be elongated to the elongation at burst during a time period smaller than the Lagrangian time macroscale. The weakest eddies that can disperse the viscous drop are thus labeled by the following multifractal exponent:(12)$$\alpha_{x}=3\frac{\ln\left\{2{\left[\frac{\beta_{\mu }C_{x}^{5/3}\mu_{D}}{\rho_{C}{\langle \varepsilon \rangle }^{1/3}L^{1/3}d}+\sqrt{{\left(\frac{\beta_{\mu }C_{x}^{5/3}\mu_{D}}{\rho_{C}{\langle \varepsilon \rangle }^{1/3}L^{1/3}d}\right)}^{2}+\frac{4C_{x}^{5/3}\sigma }{{\langle \varepsilon \rangle }^{2/3}L^{2/3}\rho_{C}d}}\right]}^{-1}\right\}}{\ln\left(\frac{L}{d}\right)}.$$

In this case, $\alpha_{x}$ depends on the interfacial tension and dispersed phase viscosity. Furthermore, the new constant, *β_μ_* (*β_μ_* = 1.91), appears. When surfactant is present in the system, an additional disruptive stress that adds to the turbulent stress given by Equation (3) may be generated. This extra stress is due to the difference between the dynamic interfacial tension of the fresh surface (exposed during drop deformation under the action of pressure fluctuations), $\sigma_{t\to 0}$, and static interfacial tension, σ [28]. This extra stress is observed when surfactant can be easily removed from the surface [28,33], but is not observed when surface active additive is strongly grafted to the surface [43]. The multifractal exponent characterizing the weakest eddies that can disperse the drop covered with surfactant, which can be removed from the surface during its deformation, is given by [28]: (13)$$\alpha_{x}=\frac{2.5\ln[(L{\langle \varepsilon \rangle }^{0.4}\rho_{C}^{0.6})/(C_{x}{\left(2\sigma -\sigma_{t\to 0}\right)}^{0.6})]}{\ln\left(\frac{L}{d}\right)}-1.5,$$
In all cases, binary breakage (number of daughter drops, $\nu \left(\upsilon^{\prime }\right)=2$) was assumed. It was also assumed that breakage into drops differing much in volume is more probable than breakage into equal drops. The daughter distribution function, $\beta \left(\upsilon ,\upsilon^{\prime }\right)$, based on the surface energy increase was used [15]. 

For comparison, a breakage model that neglects intermittency will be used. For this purpose, Coulaloglou and Tavlarides’ model [9] was chosen: (14)$$g\left(d\right)=C_{1}\frac{{\langle \varepsilon \rangle }^{1/3}}{d^{2/3}}\exp\left(-C_{2}\frac{\sigma }{\rho_{C}{\langle \varepsilon \rangle }^{2/3}d^{5/3}}\right),$$
The constants that are most often used are $C_{1}=0.00481$ and $C_{2}=0.08$ [44]. 

The coalescence rate depends on the drop collision frequency and coalescence efficiency. The average collision rate in a turbulent field is calculated using the method of steepest descent [30]. The function, $h\left(\upsilon ,\upsilon^{\prime }\right)=h\left(d,d^{\prime }\right)$ (m^3^s^−1^), appearing in the population balance equation is expressed as: (15)$$h\left(d,d^{\prime }\right)=\sqrt{\frac{8\pi }{3}}{\langle \varepsilon \rangle }^{1/3}{\left(\frac{d+d^{\prime }}{2}\right)}^{7/3}{\left(\frac{d+d^{\prime }}{2L}\right)}^{0.026}.$$
The coalescence efficiency, $\lambda \left(\upsilon ,\upsilon^{\prime }\right)=\lambda \left(d,d^{\prime }\right)$, is determined by the ratio of the average film drainage time, $t_{c}\left(d,d^{\prime }\right)$, and average interaction time $t_{i}\left(d,d^{\prime }\right)$: (16)$$\lambda \left(d,d^{\prime }\right)=\exp\left(-C\frac{t_{c}\left(d,d^{\prime }\right)}{t_{i}\left(d,d^{\prime }\right)}\right),$$
where *C* is a non-dimensional coefficient. The film drainage time depends on the mobility of drop interfaces. For pure liquid-liquid systems and a low dispersed phase viscosity, drop interfaces remain partially mobile and film drainage is controlled by the flow inside the drop. The average drainage time in intermittent turbulent flow for deformed droplets with partially mobile interfaces can be expressed as follows [30]: (17)$$t_{c}=\frac{\mu_{D}\tilde{a}R_{eq}^{2/3}}{4\sigma R_{L}^{1/2}}\left(\frac{1}{h_{c}}{\left(\frac{d_{jk}}{L}\right)}^{0.016}-\frac{1}{{\tilde{h}}_{0}}{\left(\frac{d_{jk}}{L}\right)}^{-0.01}\right).$$
The film radius, $\tilde{a}$, is derived under the assumption that the whole kinetic energy is transformed into excess surface energy, and the initial film thickness, ${\tilde{h}}_{0}$, results from a comparison of the turbulent velocity and drainage rate [29,31]. The critical (rupture) film thickness, $h_{c}$, is calculated from a comparison of the van der Waals radial force per unit volume and the pressure gradient responsible for the film thinning rate [34]. $R_{L}$ is a radius of a larger drop. The equivalent radius for unequal droplets is defined as $R_{eq}=dd^{\prime }/(d+d^{\prime })$ and $d_{jk}=(d+d^{\prime })/2$. 

The interaction time, $t_{i}$, is usually smaller than, or of the order of the time scale for two droplets to pass one another, $t_{ext}$. For intermittent turbulent flow, the average time scale, $t_{ext}$, is then given by: (18)$$t_{ext}=\frac{d_{jk}^{2/3}}{{\langle \varepsilon \rangle }^{1/3}}{\left(\frac{L}{d_{jk}}\right)}^{0.052}.$$
However, for droplets of low viscosity, the interaction time can be estimated as the time resulting from a droplet bouncing [29]:(19)$$t_{i}=\frac{1}{2}{\left(\frac{8}{3}\frac{R_{S}^{3}\left(\rho_{D}/\rho_{C}+0.75\right)\rho_{C}}{\sigma \left(1+\xi^{3}\right)}\right)}^{1/2},\;\xi =\frac{R_{S}}{R_{L}}.$$
When the dispersed phase viscosity is high the drop interfaces are immobilized. Different cases can be considered: Undeformed droplets, deformed droplets with a film radius resulting from the balance between the pressure caused by external force and Laplace pressure, and deformed droplets with a film radius proportional to the radius of the smaller droplet [31,32]. It was shown that drops of a high viscosity differing in size behave in a completely different way. In this paper, large deformed droplets and parallel-sided film are considered. In this case, the interaction time, $t_{i}=t_{ext}$, and is given by Equation (18). The film drainage controlled by Laplace pressure is assumed and the drainage time is calculated as follows [32]: (20)$$t_{c}=\frac{3\mu_{C}\rho_{C}R_{eq}^{4}{\langle \varepsilon \rangle }^{2/3}d_{jk}^{2/3}}{16\sigma^{2}h_{c}^{2}}{\left(\frac{d_{jk}}{L}\right)}^{0.026}$$

## 3. Geometry and CFD Model

The properties of turbulence for tanks equipped with one of the impellers, six-blade Rushton turbine (RT) or three blade high efficiency impeller (HE3), are determined using CFD. The impellers are shown in Figure 1. The image of HE3 is taken from [45]. Simulations are performed for a tank of a diameter, *T* = 0.15 m, and height, *H* = *T*, completely filled and closed. The stirred tank is flat bottomed and fully baffled (four equally spaced baffles of a width equal to *T*/10). It was assumed that the impeller diameter to tank diameter ratio is *D*/*T* = 0.4 and the impeller clearance is *C*/*T* = 1/4 for HE3. The high efficiency impeller has a uniform blade width equal to 0.01 m. The blade angle is 30 degrees at the hub. The tip chord angle is 15 degrees. The blade is bent at 50% of its length. The thickness of the blade is equal to 0.002 m. The Rushton turbine has a diameter of *D* = 0.5*T*. A disc diameter is equal to 0.75 *D*, a blade thickness and disk thickness are 0.01 *D*. The impeller clearance is *C*/*T* = 1/2.

The unstructured tetrahedral meshes with approximately 400,000 cells for a tank equipped with a Rushton turbine and 600,000 cells for a high efficiency impeller were generated using Mixsim software. Steady state 3D simulations were performed using the finite volume package, Ansys Fluent. The multiple reference frame approach and standard *k*-*ε* model with standard wall functions were used. The SIMPLE algorithm was used for pressure-velocity coupling. The PRESTO scheme was used for pressure interpolation, and the second order upwind scheme was used for the momentum, kinetic energy, and energy dissipation rate equations.

The CFD simulations were performed to obtain power numbers, *P_o_*, pumping capacity (flow number, *Fl*), as well as the normalized mean energy dissipation rate and normalized integral scale of turbulence in the impeller and bulk zones ($\varphi_{imp}$, $\varphi_{bulk}$, $L_{imp}/D$, $L_{bulk}/D$). These values were then used in the circulation flow model (the dispersion circulates through the impeller and bulk zones). A multifractal model allows the probability of stresses characterized by different multifractal exponents, α, for a given average energy dissipation rate calculated for a given zone to be predicted. Such a model gives excellent results in predicting drop size evolution as was shown in previous papers [30,33,43]. The details of the flow pattern were not used. 

## 4. Results and Discussion

CFD simulations were performed for a high efficiency impeller (HE3) for an impeller speed of *N* = 800 rpm. The operating fluid was water. The presence of an organic phase was not taken into account in these simulations. It was justified by low values of the dispersed phase volume fraction (*φ* = 0.001 for the pure breakage case and *φ* = 0.05 for the coalescence case). For the tank equipped with a Rushton turbine, simulations were performed for the impeller speed that was expected to give the same power input per unit mass (*N* = 213 rpm). These simulations allowed the power and flow numbers for both impellers to be determined: *P_o_* = 0.34, *Fl* = 0.39 for the high efficiency impeller, and *P_o_* = 4.98, *Fl* = 0.74 for the Rushton turbine. These values agree well with the measured ones (*P_o_* = 0.305 and *Fl* = 0.41 for HE3 of D/T = 0.46 [46], *P_o_* = 0.3 for HE3 of D/T = 0.39 and an impeller clearance equal to T/4 [40]). Power number values for RT reported by different authors are in the range of 4.6 to 6.3. According to Bujalski et al., the correlation based on experimental measurements of the power number, *P_o_*, depends on the tank size and the impeller disc thickness [47]. For *T* = 0.15 m and a disc thickness equal to 0.01 *D,* the power number should be equal to 5.5. However, this correlation was obtained for vessels of a diameter from 0.22 m to 1.83 m. The flow generated by HE3 at the clearance of *T*/4 has a strong axial component directed to the base. Between the impeller hub and the tank base, there is a weak reverse flow. Local values of the integral scale, *L*, were determined using calculated local values of the energy dissipation rate, *ε*, and turbulent kinetic energy, *k*, *L* = (2*k*/3)^3/2^/*ε*. The contours of *L* for both types of impellers are presented in Figure 2. In both cases, there is a distinct difference in the integral length scale in the impeller zone and the bulk, though the average values of *L* in these zones do not differ as much as the values of the energy dissipation rate do.

Earlier studies of the author [48,49] have shown that the multiple zone model of the tank (10-zones) predicts similar drop size distributions as the 2-zone model, provided that the impeller zone is properly defined. For example, for the Rushton turbine of *D*/*T* = 1/3, only part of the impeller stream should be included into the impeller zone. However, in the case of RT of D/T = 1/2 that is considered in the present paper, the whole discharge region is included to the impeller zone. This impeller stream region together with the impeller swept volume (extended here 3 mm above and 3 mm below impeller blades) occupies a fraction, *x_imp_* = 0.095, of the tank volume. To determine the mean values of the energy dissipation rate and the scale of large eddies, auxiliary surfaces were created and the surface integrals function was used. Surfaces were separated by 1 mm in the impeller zone. Such densely created surfaces enabled us to better define the limits of the impeller zone. For the bulk zone, the surfaces were separated by 2 mm. The relative properties of the turbulence for the Rushton turbine of D/T=1/2 are as follows: $\varphi_{imp}={\langle \varepsilon \rangle }_{imp}/\bar{\langle \varepsilon \rangle }=6.1$, $\varphi_{bulk}=0.465$, $L_{imp}/D=0.0806$, $L_{bulk}/D=0.13$, where $\bar{\langle \varepsilon \rangle }=\rho P_{o}N^{3}D^{5}/(\rho V)$. In the case of HE3, the impeller zone is defined as a cylinder of a radius of *r* = 0.034 m (slightly larger than the impeller radius, *R* = 0.03 m) and positioned between *z* = 0.0325 m and *z* = 0.0425 m. The volume fraction of this zone is equal to *x_imp_* = 0.0137, the normalized mean energy dissipation rate in this zone is $\varphi_{imp}=45$, while in the bulk it is $\varphi_{bulk}=0.389$. The normalized mean integral scales of the turbulence are $L_{imp}/D=0.0573$ in the impeller zone, and $L_{bulk}/D=0.187$ in the bulk. The spatial distribution of the energy dissipation rate in the impeller zone has been reported for the RT by many researchers. However, most works were devoted to impellers of a diameter of *D* = *T*/3. The percentage of the total energy dissipated in the swept impeller and the impeller stream regions reported by different authors vary from 42% to 70% [50]. For an impeller diameter of *D* = *T*/2, the normalized energy dissipation rate in the impeller zone is $\varphi_{imp}=5.93$ and the impeller volume fraction is $x_{imp}=0.105$ according to the Okamoto correlation [51]. Thus, the percentage of the total energy dissipation in the impeller zone is 62.3%. Zhou and Kresta [52] measured a 43.5% dissipation in the control volume containing impeller swept and impeller stream zones and occupying 10% of the tank volume. The percentage of the total energy dissipation in the impeller region predicted in the present work (57.95%) lies between these literature values. The used *k*-*ε* model gives reasonable results. The Reynolds stress model (RSM) used in previous work [48] was able to predict the characteristic properties of the energy dissipation rate profiles. For example, it predicts that the rate of the energy dissipation in the impeller stream for a radial position of r/R = 1.325 is much higher than at smaller and larger radial distances. It agrees with the PIV measurements of Baldi and Yianneskis [53], who observed a similar jump at a radial position of r/R = 1.32. The *k*-*ε* model does not predict any jump. It predicts the decrease of *ε* with the increase of the distance from the impeller blades. However, the normalized mean energy dissipation rate in the impeller stream predicted by both turbulence models differs only by 2%. The difference between the pumping capacity predicted by both models was smaller than 1.4%.

The population balance equation was solved for three liquid-liquid systems. In the first case, the dispersed phase of low viscosity is considered ($\mu_{D}=0.001$ Pa·s). The density of the continuous phase is assumed to be $\rho_{C}=1000$ kg/m^3^, and the interfacial tension is σ=0.035 N/m. The second liquid-liquid system is characterized by $\mu_{D}=0.5$ Pa·s, $\rho_{C}=1000$ kg/m^3^, and σ=0.035 N/m. For both these liquid-liquid systems, pure breakage (dispersed phase volume fraction, φ=0.001) as well as breakage together with coalescence (φ=0.05) were simulated. In the calculations, the constant, *C*, in Equation (16) defining the coalescence efficiency is assumed to be *C* = 0.5 for the first liquid-liquid system characterized by partially mobile interfaces and drainage and interaction times are calculated from Equations (17) and (19), respectively. For the system with a drop surface mobility decreased due to a high dispersed phase viscosity, Equations (20) and (18) are used to estimate the drainage and interaction times and the constant is equal to *C* = 0.1. The Hamaker constant, which influences the critical film thickness, is assumed to be *A* = 10^−20^ J (characteristic for pure liquid-liquid systems). The third liquid-liquid system contains surfactant, which is easily removed from the surface. Because of the surfactant presence, the coalescence is not observed when starting from big droplets (very slow coalescence could be observed after an impeller speed reduction—see [28,33]), and additional disruptive stress appears due to the interfacial tension difference between the freshly exposed interface and the interface covered by surfactant [28]. In this case, a multifractal exponent characterizing the weakest eddies is calculated from Equation (13). The interfacial tension values are σ=0.0233 N/m and $\sigma_{t\to 0}=0.0255$ N/m (as measured for toluene/1 mM sodium dodecyl sulfate SDS aqueous solution [33]). An initial drop size of *d* = 3 mm was assumed in the calculations. 

Figure 3 shows the transient drop size distributions predicted for both types of impeller at the same power input per unit mass for conditions when only the breakage of droplets of a low viscosity (liquid-liquid system 1) takes place. The drop size distribution at short agitation times is much wider for the HE3. However, the mean Sauter diameter, *d*_32_, is only slightly larger for HE3 than for RT. This is because the largest volume fraction of drops is formulated by smaller droplets in the tank equipped with the HE3 than in that equipped with the RT. The higher the power input per unit mass (and thus the higher mean energy dissipation in the tank), the smaller the observed *d*_32_ for the HE3 in comparison with *d*_32_ for the RT (even after a few minutes of agitation). However, the largest droplets are still bigger for the HE3 than for the RT at short agitation times. After long agitation times, droplets produced by the HE3 are much smaller than droplets produced by the RT.

A comparison of the drop size distributions and mean sizes of drops produced by different impellers after 2 h of agitation is shown in Figure 4. The described behavior of droplets can be explained by the smaller impeller zone volume and larger $\varphi_{imp}$ in the tank equipped with a high efficiency impeller. Very large droplets that are easily broken in both systems have a greater chance of appearing in the zone of the high energy dissipation rate and, therefore, high turbulent disruptive stresses in the tank with the RT ($x_{imp,RT}>x_{imp,HE3}$). However, the final drop size is determined by the magnitude of ${\langle \varepsilon \rangle }_{imp}$ and this is much higher for the HE3. 

Breakage and coalescence models were previously verified experimentally for Rushton turbines of different *D*/*T* ratios. A comparison between the measured (literature as well as our own experiments) and the predicted distributions (with the energy dissipation rate distribution based on the experimental Okamoto correlation as well as being predicted using the CFD) one can find elsewhere [2,30,31,32,33,54]. Some experimental results from the literature (for long agitation times) for the RT, *D* = *T*/2, for low as well as high dispersed phase viscosity [13,54,55,56,57] are presented in Figure 4b and Figure 6b. In the case of Arai et al.’s [55] experiments, it was assumed that the Sauter diameter is $d_{32}=0.6d_{\max}$ ($d_{\max}$ is the diameter of the maximum stable drop size) for a dispersed phase of low viscosity and $d_{32}=0.5d_{\max}$ for a dispersed phase of high viscosity. When the power number was not measured, the value of *P_o_* = 4.98 was used to estimate the power input per unit mass. Additional information is shown in Figure 4b and Figure 7b. Good agreement between the model predictions and experimental data was obtained. The slopes of the lines in Figure 4b, obtained by solving the PBE with the multifractal breakage model derived for the dispersed phase of low viscosity (Equations (9) and (11)), are −0.535 for the Rushton turbine and −0.547 for the high efficiency impeller, respectively. These values are closer to the exponent of −0.617 resulting from Equation (5) for the minimum multifractal exponent, $\alpha =\alpha_{\min}=0.12$ (characterizing the most vigorous turbulent events and highest stresses), than to the −0.4 predicted for the most probable events characterized by α≈1 (see Equation (5) for α=1). A slope of −0.4 is also predicted when intermittency is neglected and no multifractal exponent is introduced. The limiting value of the maximum stable drop size, $d_{\max}\propto \varepsilon^{-0.617}$, corresponds to $d_{\max}\propto We^{-0.93}$. Such a low exponent on the Weber number was first observed by Konno and Saito [58]. The exact value of this exponent is predicted by the multifractal breakage model. As was shown in previous papers, the multifractal breakage model not only predicts this exponent, but it also predicts transient drop size distributions very well [2,28,43], both for short agitation times, when the most probable stresses determine the drop size, and for long agitation times, when droplets of a diameter, *d*, are not disrupted by stresses, p(d,α=1), but can be broken by stresses characterized by α<1. The really stable droplet size is given by $d_{32}\propto \varepsilon^{-0.617}$ (because the Sauter diameter, *d*_32_, is proportional to *d*_max_). The dashed line in Figure 4b shows the Sauter diameter predicted for the Rushton turbine by Coulaloglou and Tavlarides’ model [9], which does not take into account local intermittency. In this case, the slope of the line is close to −0.4. Figure 5a shows a comparison of the transient drop size distributions predicted by Coulaloglou and Tavlarides’ model for the Rushton turbine and high efficiency impeller for short (*t* = 10 min) and long (*t* = 2 h) agitation times. This model also predicts that the HE3 produces broader DSD at short agitation times and smaller drops at long agitation times. Figure 5b shows the distributions predicted for the Rushton turbine by different models. One can see that Coulaloglou and Tavlarides’ model predicts smaller droplets. It can also be observed in Figure 4b.

Similar trends as those shown in Figure 3 and Figure 4 are observed for the breakage of droplets of high viscosity, Figure 6 and Figure 7, i.e., smaller droplets produced by the HE3 impeller. Calculations were performed using Equations (9) and (12). The slopes in Figure 7b are −0.365 for the RT and −0.354 for the HE3. As can be seen from Equation (6), the dependence of *d*_max_ on the energy dissipation rate is more complicated than for low viscosity, but again the limiting size is determined by $\alpha_{\min}=0.12$. The slope for the case when intermittency is neglected can be predicted by setting α=1. In this case, ${\left(d_{\max}/L\right)}^{0.4\left(1-\alpha \right)}$ and ${\left(d_{\max}/L\right)}^{\left(1-\alpha \right)/3}$ disappear. Additionally, when 1 is small in comparison with the second term in a square bracket (i.e., when the effect of the shape restoring stress due to interfacial tension is small in comparison with the viscous stabilizing stress), the resulting slope can be estimated as −0.25. Thus, again, −0.365 (or −0.354) is smaller than −0.25, which shows that intermittency is important.

Figure 8 shows transient drop size distributions predicted for droplets stabilized by surfactant when an additional disruptive stress due to the interfacial tension difference is generated (multifractal exponent, $\alpha_{x}$, is calculated from Equation (13)). Again, smaller droplets are produced in the tank equipped with a high efficiency impeller. 

Figure 9a presents the final drop size distributions (being the result of dynamic equilibrium between breakage and coalescence) produced by different impellers at a higher dispersed phase volume fraction (*φ* = 0.05) for a pure liquid-liquid system (no surfactant) with a dispersed phase of low viscosity. Under such conditions, droplets have partially mobile interfaces and gentle collisions are favored (Equations (17) and (19)). Therefore, fast coalescence takes place in the bulk. The model predicts that at the same power input per unit mass, droplets produced in both systems are similar. In the case of a high dispersed phase viscosity, coalescence is highly hindered by immobilization of the drop interfaces. However, very wide drop size distributions are usually observed for such dispersions and as was shown earlier, droplets differing in size or in deformation in the contact area behave in a completely different way [32]. Coalescence of small rigid drops or small slightly deformed drops may be even faster in the zone of the high energy dissipation rate than in the bulk. However, the largest volume fraction of the population (being the result of dynamic equilibrium between breakage and coalescence) is occupied by large deformed droplets. Their behavior can be predicted using Equations (18) and (20). The predicted drop size distributions are shown in Figure 9b. Slightly smaller droplets are produced by the HE3. 

From the presented results, it follows that smaller droplets are produced at the same power input per unit mass by a low power number, high efficiency impeller (HE3) when drop breakage prevails.

## Figures and Tables

**Figure 1 entropy-21-00340-f001:**
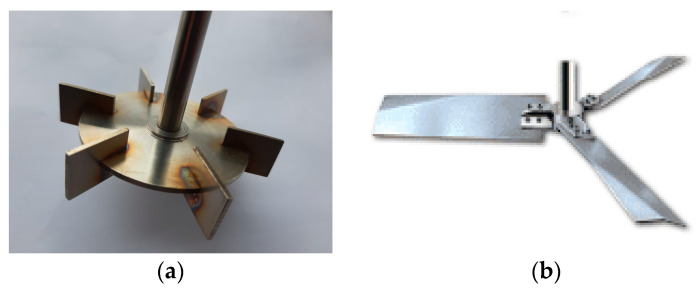
Impellers: (**a**) Rushton turbine and (**b**) high efficiency impeller.

**Figure 2 entropy-21-00340-f002:**
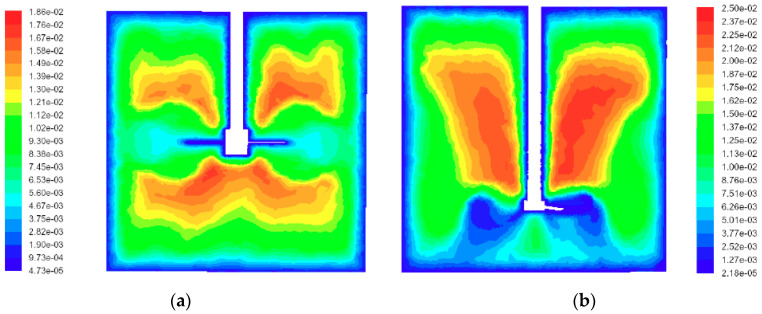
Contours of the integral scale for: (**a**) Rushton turbine and (**b**) HE3 impeller (plane θ = 45^°^ between baffles).

**Figure 3 entropy-21-00340-f003:**
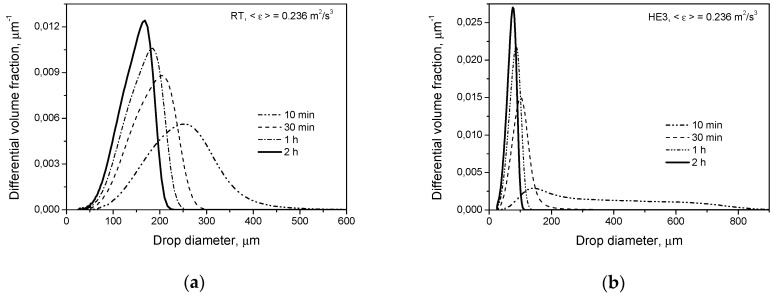
Influence of the impeller type on transient drop size distributions for a dilute system (*φ* = 0.001) with a dispersed phase of low viscosity ($\mu_{D}=0.001$ Pa·s): (**a**) Rushton turbine; (**b**) high efficiency impeller, HE3.

**Figure 4 entropy-21-00340-f004:**
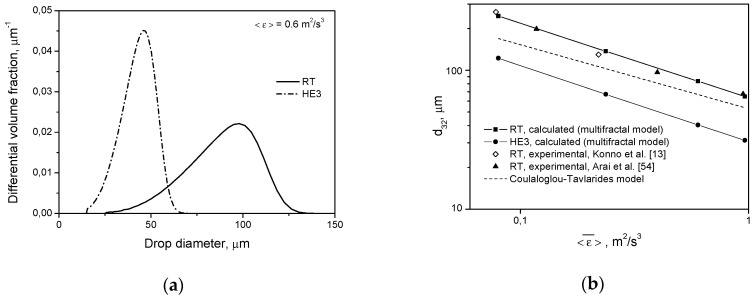
Influence of the impeller type on: (**a**) drop size distribution at *t* = 2 h (pure breakage, $\mu_{D}=0.001$ Pa·s, *φ* = 0.001); (**b**) Sauter diameter.

**Figure 5 entropy-21-00340-f005:**
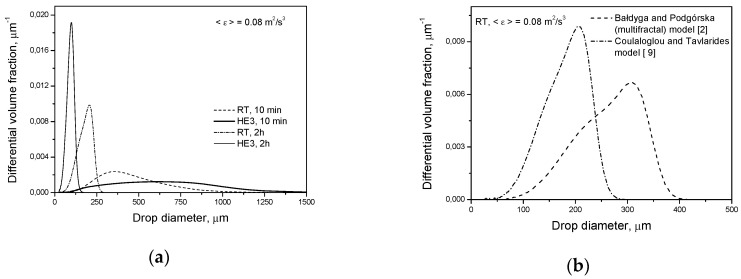
Drop size distributions: (**a**) influence of the impeller type on the transient drop size distributions predicted by Coulaloglou and Tavlarides’ model; (**b**) drop size distributions at *t* = 2 h (pure breakage, $\mu_{D}=0.001$ Pa·s, *φ* = 0.001)—comparison of models.

**Figure 6 entropy-21-00340-f006:**
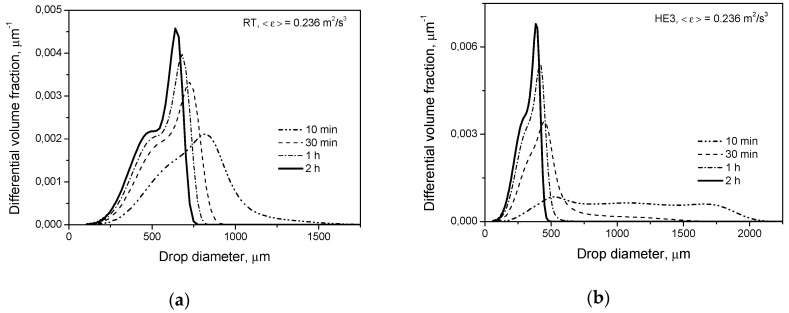
Influence of the impeller type on transient drop size distributions for a dilute system (*φ* = 0.001) with a dispersed phase of high viscosity ($\mu_{D}=0.5$ Pa·s): (**a**) Rushton turbine; (**b**) high efficiency impeller.

**Figure 7 entropy-21-00340-f007:**
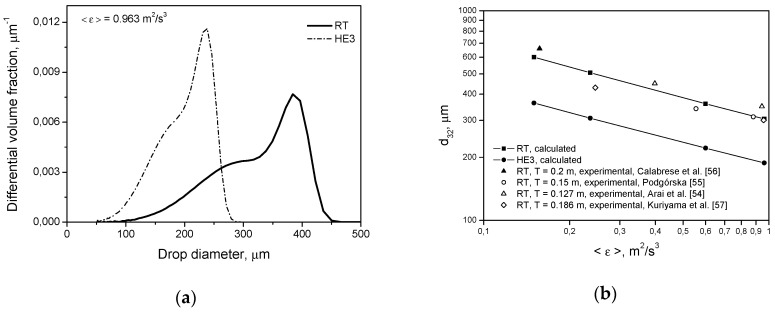
Influence of the impeller type on: (**a**) drop size distribution at *t* = 2 h (pure breakage, $\mu_{D}=0.5$ Pa·s, *φ* = 0.001); (**b**) Sauter diameter.

**Figure 8 entropy-21-00340-f008:**
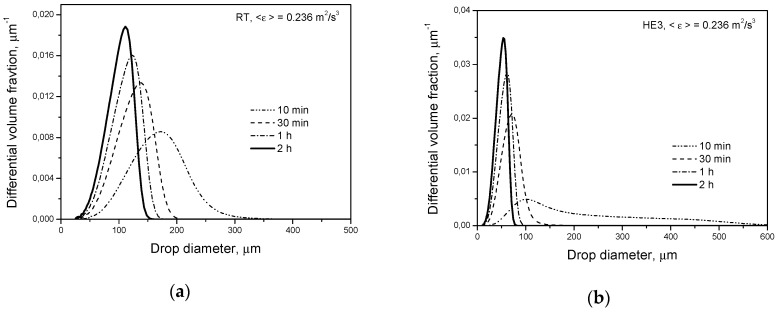
Influence of the impeller type on transient drop size distributions for a dilute system (*φ* = 0.001) with droplets covered by surfactant: (**a**) Rushton turbine; (**b**) high efficiency impeller.

**Figure 9 entropy-21-00340-f009:**
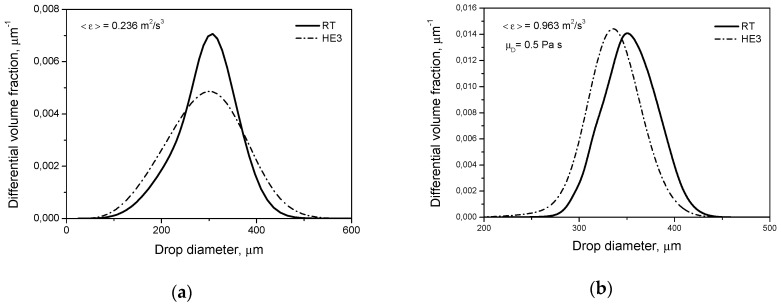
Influence of the impeller type on the drop size distribution in: (**a**) fast coalescing dispersion ($\mu_{D}=0.001$ Pa·s, *φ* = 0.05); (**b**) slowly coalescing dispersion ($\mu_{D}=0.5$ Pa·s, *φ* = 0.05).

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
