# Peer review of "The Influence of Internal Intermittency, Large Scale Inhomogeneity, and Impeller Type on Drop Size Distribution in Turbulent Liquid-Liquid Dispersions"

_entropy, 2019, doi:10.3390/e21040340_

Round 1

Reviewer 1 Report

1) Paper is written in proper English, but some editorial work is needed-see the correction marked in the scanned and attached page 1.

14-and also properties,17-when the only breakage..,with a dispersed...,19-atsame power,18...,as wel las high viscosity,... 21-..of the dispersed.., ...when the mobility...,27- in a turbulent, 28-...pharmaceutical, and...29-..of a heterogeneous chemical reaction..., 33-...also determinates ...,34..i.e, the process,...36-...intermittency also affects the...

2) Description of the impeller geometry is rather fussy. I would recommend giving the drawing and reference showing the impeller geometry and its origin.
3) Page8 and 9: Are there some other models describing the dispersion time course?

4) What are the limiting (equilibrium) values of d32 and PSD curve for the infinite time according to the model? Are the results for the infinite time in accordance with practical experience?

5) What are the slopes of the straight lines in Figures 3b and 4b)? How close are they to

the expected  value -0.4?

Author Response

Please check our responses in the attachment.

Reviewer 2 Report

This article studies the influence of impeller type on drop size distribution in an stirred vessel. The effect of the intermittency of the turbulence on droplet breakage and coalescence is taken into account by means of a multifractal formalism. The paper is very interesting and in my opinion it should be published in Entropy.

In general, the paper is well written and the English is correct. However, some minor spelling mistakes have been detected (they are highlighted in yellow in the attached comented version of the article).

I have the following comments:

·         30% of all references are from the author

·         The employed grid seems not to be very refined for the 3D problem. Nothing is said about the grid convergence study. Can the author tell something at this respect?

·         Details about the CFD computation should be provided: order of discretization schemes (spatial and temporal), pressure-velocity coupling employed, size of time step

·         Probably the k-epsilon model is not the most appropriate model for simulating the stirred vessel. Have the author tried some other turbulence models?

·         It would be interesting for the reader to have a comparison of results, at least in one case, when the turbulence intermittency is not considered

Author Response

(The authors gave the same response as above.)
